# The Physiologic Significance of Early Urinary Intestinal Fatty Acid Binding Protein Levels in Preterm Infants: A Prospective Cohort Study

**DOI:** 10.3390/children8100842

**Published:** 2021-09-24

**Authors:** Young-Hwa Jung, Ee-Kyung Kim, Seung-Han Shin, Jin-A Lee, Han-Suk Kim, Beyong-Il Kim

**Affiliations:** 1Department of Pediatrics, Seoul National University Bundang Hospital, Seoul National University College of Medicine, Seongnam-si 13620, Korea; jyhtlcn@gmail.com (Y.-H.J.); 66105@snubh.org (B.-I.K.); 2Department of Pediatrics, Seoul National University Children’s Hospital, Seoul National University College of Medicine, Seoul 03080, Korea; revival421@snu.ac.kr (S.-H.S.); kimhans@snu.ac.kr (H.-S.K.); 3Department of Pediatrics, Seoul National University Boramae Medical Center, Seoul National University College of Medicine, Seoul 07061, Korea; ljinna@snu.ac.kr

**Keywords:** intestinal fatty acid binding protein, fetal intestine, perinatal distress, intestinal morbidity, premature infant, feeding intolerance

## Abstract

Intestinal fatty acid binding protein (I-FABP) is released from mature enterocytes when cell membrane integrity is disrupted. This study aimed to prospectively investigate the physiologic significance of early urinary I-FABP and whether it might reflect intestinal compromise in preterm infants. We conducted a prospective cohort study of 100 preterm infants weighing <1250 g and collected serial urine samples at 12, 24, and 48 h after birth. The correlations between initial urinary I-FABP/urinary creatinine (creatinine_u_) levels and associated factors were analyzed. Among 100 patients, 15 were diagnosed with meconium obstruction of prematurity, and five were diagnosed with necrotizing enterocolitis during the hospital stay. Early urinary I-FABP/creatinine_u_ levels were inversely correlated with both gestational age (Spearman’s rank correlation coefficient (Rs) −0.381, *p* < 0.01) and birth weight ((Rs) −0.424, *p* < 0.01). Early urinary I-FABP/creatinine_u_ levels were associated with cord pH ((Rs) −0.436, *p* < 0.01) and base excess ((Rs) −0.258, *p* = 0.021). There were significantly positive correlations between early urinary I-FABP/creatinine_u_ levels and the time to full enteral feeding in preterm infants without specific intestinal morbidities. Therefore, a more premature gut with acute perinatal ischemia is expected to exhibit increased I-FABP levels shortly after birth. Because of small sample size, further large-scale studies are needed.

## 1. Introduction

Intestinal fatty acid binding protein (I-FABP) comprises a 15-kDa cytosolic water-soluble protein located in the small and large intestines and is released when the intestinal mucosa is disrupted [1]. I-FABP can be filtered via glomeruli because of its low molecular weight and can be measured in urine quickly. Several recent studies have reported that I-FABP, which reflects mucosal compromise, can be used as a promising biomarker of acute intestinal mucosa damage, including necrotizing enterocolitis (NEC) in preterm infants, and assesses the extent and severity of intestinal involvement even before the development of disease [2,3,4,5,6].

In the fetus, the intestine is susceptible to blood flow redistribution, especially that associated with fetal intestinal hypoperfusion caused by placental insufficiency [7,8]. Few studies have demonstrated that a decreased intrauterine vascular supply to the gastrointestinal tract predisposes patients with fetal growth retardation to NEC development [9,10]. In addition, chorioamnionitis may contribute to fetal gut immune alterations through activation of the maternal innate immune system, which activates a strong fetal inflammatory response [11,12].

Therefore, the objective of this prospective study was to evaluate the physiologic significance of I-FABP levels shortly after birth and to assess whether they are associated with factors related to perinatal intestinal hypoperfusion or fetal inflammation. We also investigated the possible associations of early I-FABP levels with postnatal intestinal outcomes.

## 2. Materials and Methods

### 2.1. Study Design

This prospective cohort study included 100 preterm infants with birth weights < 1250 g who were admitted to the neonatal intensive care unit of Seoul National University Children’s Hospital between October 2012 and September 2015. Patients with major congenital anomalies such as cyanotic heart diseases, ductus dependent heart disease, abdominal congenital anomalies, neural tube defect, and chromosomal anomalies, were excluded.

### 2.2. Sample Acquisition and Assay Procedure

Urine was collected for 6 h by placing a cotton ball near the urethral meatus within the diaper to avoid stool contamination. Once the cotton ball was saturated, urine was obtained by squeezing it into a sterile syringe and collected in a urine tube. The samples were centrifuged, and the supernatants were frozen at −80 °C until evaluation [13]. The urinary concentration of I-FABP was quantified in duplicate samples using a commercially available enzyme-linked immunosorbent assay (HK406; Hycult Biotech, Plymouth Meeting, PA, USA). The urinary creatinine concentration (creatinine_u_) was determined in duplicate samples using a colorimetric assay (Cayman Chemical Co., Ann Arbor, MI, USA). Urine was analyzed for creatinine content to calculate the I-FABP/creatinine_u_ ratio.

### 2.3. Sample Collection Protocol

Initial urine samples were collected at 12 ± 6, 24 ± 6, and 48 ± 6 h after birth from all participants. Early urinary I-FABP/urinary creatinine (creatinine_u_) was defined as the mean value of initial urinary I-FABP/creatinine_u_ levels for the first 48 h after birth. Urine was re-collected at 12 ± 6, 24 ± 6, and 48 ± 6 h after the onset of symptoms of meconium obstruction of prematurity (MOP) or NEC.

### 2.4. Group Allocations

At the first presentation of symptoms, at the discretion of the clinical teams, the patients were allocated to the NEC group or the MOP group, and urine was collected according to the study protocol. The final group was reassigned when the study was completed.

### 2.5. Diagnosis of MOP

MOP was suspected when infants had difficulty with meconium passage and presented with abdominal distension, bilious gastric residue and feeding intolerance. A plain X-ray of the abdomen showed dilated bowel loops without pneumatosis intestinalis, and ultrasonography findings included meconium-filled dilated bowel loops without evidence of NEC. The diagnosis was finally confirmed by intervention resulting in the passage of meconium plugs and the resolution of intestinal obstruction [14]. Gastrografin (Bayer plc, Newbury, Berkshire, UK) enemas were used in the diagnosis and management of MOP.

### 2.6. Diagnosis of NEC

NEC was defined as the presence of pneumatosis intestinalis on abdominal X-ray corresponding to Bell’s stage II or higher.

### 2.7. Feeding Protocol

During the study period, the same nutritional protocol was applied by the same assigned neonatologists. Enteral feeding was started via an orogastric tube as an intermittent bolus every 3 h, when the infants were stable, mostly within 24 h of life. The initial feeding volume was 10–20 mL/kg/day, and when it was tolerated, this volume was gradually advanced by 10–20 mL/kg/day. The days to reach full enteral feeds were defined according to a volume of >120 mL/kg/day. If enteral intolerance or medical instability was noted, the volume was reduced or withheld. Enteral intolerance was as follows: the inability to digest enteral feeding presenting as a gastric residual volume of more than 50%, abdominal distension with a visible bowel loop, and bile-stained aspirates or emesis or both [15].

### 2.8. Outcome Assessment

All data were prospectively collected and included demographics, other clinical data, maternal data, and disease characteristics. We assessed the correlations between early urinary I-FABP/creatinine_u_ levels during 48 h after birth and prenatal and postnatal factors. We evaluated the correlations between early urinary I-FABP/creatinine_u_ levels and the time required to achieve full enteral feeding in the normal group, which included preterm infants without any intestinal morbidities such as MOP or NEC.

### 2.9. Statistical Analysis

Normality was assessed by the Shapiro–Wilk test. The Mann–Whitney U test and the Kruskal–Wallis test were used to analyze continuous variables, the χ^2^ test and Fisher’s exact test were used for categorical variables, and Spearman’s rank test was used for correlations. Significant variables (*p* < 0.05) identified on univariate analyses were entered into the regression model. We used multiple linear regression analyses to determine which perinatal variables were independently associated with urinary I-FABP/creatinine_u_ levels. Statistical analysis was performed using SAS 9.4 (SAS Institute, Cary, NC, USA) and STATA^®^ 14 (Stata Corporation, College Station, TX, USA).

### 2.10. Study Approval

The Institutional Review Board at Seoul National University Children’s Hospital approved the study, and written informed consent was obtained from the parents of participants prior to inclusion in the study. Research was performed in accordance with the Declaration of Helsinki (ethical principles for medical research involving human subjects). This trial has been registered at www.clinicalTrials.gov (accessed 12 August 2016, NCT02864446).

## 3. Results

### 3.1. Patient Clinical Characteristics

A total of 100 preterm infants weighing <1250 g were enrolled during the study period. Nine patients died: four died within 7 days after birth, none developed NEC or meconium obstruction of prematurity (MOP), and one patient achieved full enteral feeding. Finally, five patients were diagnosed with stage ≥2 NEC, and 15 patients were diagnosed with MOP (Figure 1).

The demographic and clinical characteristics of the study population are presented in Table 1.

The mean I-FABP/creatinine_u_ levels at 12 h, 24 h, and 48 h after birth were 5.25 ± 10.18 pg/nmol, 5.34 ± 7.53 pg/nmol, and 5.20 ± 7.74 pg/nmol, respectively. The early I-FABP/creatinine_u_ levels were not significantly different among the normal group, the MOP group, and the NEC group. (Data not shown)

### 3.2. Early I-FABP/Creatinine_u_ Levels and Associated Factors

Initial I-FABP/creatinine_u_ levels were inversely correlated with gestational age and birth weight (Figure 2).

The 1-min and 5-min Apgar scores were not correlated with early urine I-FABP/creatinine_u_ levels. However, umbilical cord pH and base excess had significant inverse relationships with early urine I-FABP/creatinine_u_ (Table 2).

Table 3 displays the associations between early I-FABP/creatinine_u_ levels and prenatal and postnatal factors in preterm infants.

There were no significant correlations with prenatal factors, such as small for gestational age, pregnancy-induced hypertension, histologic chorioamnionitis, funisitis, premature rupture of membrane, and administration of antenatal steroids. Only oligohydramnios was significantly associated with early I-FABP/creatinine_u_ levels (*p* = 0.003). Early urinary I-FABP/creatinine_u_ levels were significantly associated with bronchopulmonary dysplasia (BPD) (*p* = 0.023) and severe retinopathy of prematurity (ROP) (*p* = 0.025). Multiple linear regression analysis demonstrated that younger gestational age, lower birth weight, and lower cord pH were the most consistent risk factors for increased levels of early FABP/creatinine_u_ (Table 4).

### 3.3. Early I-FABP/Creatinine_u_ and Feeding Tolerance in the Normal Group

The median duration to achieve full enteral feeding was 14 days (3–33 days), and the median Nil per os (NPO) duration was 1 day (0–12 days). The median proportion of NPO duration to the time required to achieve full enteral feeding was 8.8% (0–55%). There were significant positive linear relationships between the time until full enteral feeding and early I-FABP/creatinine_u_ levels ((Rs) 0.442, *p* = 0.001). In addition, there were significant correlations between early I-FABP/creatinine_u_ levels and the duration of NPO ((Rs) 0.311, *p* = 0.008). However, after adjusting for gestational age, the correlations between the time until full enteral feeding and early I-FABP/creatinine_u_ levels decreased ((Rs) 0.295, *p* = 0.016).

### 3.4. Comparisons of Early Urinary I-FABP/Creatinine_u_ Levels between the Normal Group and the MOP Group or the NEC Group

Fifteen patients were confirmed to have MOP at a median age of 3 days (1–6 days) after birth. Early I-FABP/creatinine_u_ levels were not significantly different between the normal group and the MOP group. When we compared the I-FABP/creatinine_u_ levels at different time points, the I-FABP/creatinine_u_ levels at 12 h and 24 h after birth were not significantly different from those of the normal group, but the I-FABP/creatinine_u_ level at 48 h after birth was significantly higher in the MOP group (5.65 ± 8.34 pg/nmol vs. 8.95 ± 7.05 pg/nmol, *p* = 0.025) (Figure 3).

Early I-FABP/creatinine_u_ levels and the I-FABP/creatinine_u_ levels at 12 h, 24 h, and 48 h after birth in the NEC group were not significantly different from those in the normal group.

## 4. Discussion

We investigated the physiological significance of early urinary I-FABP levels for the first 48 h after birth in preterm infants. To the best of our knowledge, this is the first study to evaluate early urinary I-FABP levels in preterm infants.

To date, several studies have suggested that urinary I-FABP levels in preterm infants are correlated with intestinal mucosal injury, as occurs in NEC. However, there is still a lack of data about the physiologic significance of urinary I-FABP levels in preterm infants.

Fetal intestinal compromise can be presumably associated with intestinal ischemia caused by blood flow redistribution due to fetal systemic hypoperfusion and intestinal inflammation caused by chorioamnionitis. As a consequence, infants born with fetal intestinal compromise are thought to have impaired gut function after birth, which may result in intestinal disturbances, ranging from temporary intolerance of enteral feeding to full blown NEC [8,9,16].

Several studies have shown that the serum I-FABP level is elevated in adult patients with intestinal ischemia [17,18]. In the fetus, intestinal ischemia can occur when fetal systemic perfusion is compromised. In this study, lower umbilical cord pH values were significantly correlated with higher early urinary I-FABP/creatinine_u_ levels even after adjusting for gestational age. However, prenatal factors that signify more sustained hypoperfusion conditions, such as preeclampsia and fetal growth restriction, were not correlated with early urinary I-FABP/creatinine_u_ levels. This might suggest that early urinary I-FABP levels are more representative of the acute phase of intestinal ischemia than chronic perturbation.

We also investigated the associations between intestinal injury caused by inflammation and early urinary I-FABP levels. In the present study, prenatal factors associated with fetal inflammation, including premature rupture of the membrane, histologic chorioamnionitis, and funisitis, were not correlated with early urinary I-FABP/creatinine_u_ levels. A few studies have evaluated I-FABP as a predictive marker of intestinal injury in patients with inflammatory bowel disease. To date, the role of I-FABP in predicting disease activity or extent in inflammatory intestinal injury remains controversial. A few studies showed higher I-FABP levels in active ulcerative colitis patients than in healthy controls [19]. In addition, Wiercinska-Drapalo A et al. demonstrated that elevated serum I-FABP concentrations in patients with ulcerative colitis were correlated with an extended inflammatory process [17,19]. However, Bodelier AG et al. reported that plasma I-FABP did not differ between endoscopically active disease and remission in inflammatory bowel disease [20]. According to our results, early urinary I-FABP levels might be a candidate biomarker for indicating intestinal ischemia, but the way fetal inflammatory status affects these levels cannot be deduced from our study.

Feeding intolerance is extremely common in premature infants and is mainly attributed to functional immaturity of the gastrointestinal tract, which affects the motility and secretion of gut hormones [15]. In addition, underlying causes that involve perinatal intestinal hypoperfusion also affect gastrointestinal dysmotility in preterm infants [21,22]. In this context, we evaluated whether early urinary I-FABP/creatinine_u_ levels can predict enteral feeding prognosis in preterm infants without any specific intestinal morbidities. Our results showed significant correlations between urinary I-FABP/creatinine_u_ levels during the first 48 h after birth and the time required to achieve full enteral feeds and the duration of NPO. However, these correlations decreased after adjusting for gestational age in preterm infants. Instead, younger gestational age and lighter birth weight were significantly associated with increased levels of early urinary I-FABP/creatinine_u_ in preterm infants. Reisinger et al. reported that increased gestational age was accompanied by an increase in the tissue content of I-FABP in sheep and humans [23]; however, they also showed that human urinary I-FABP levels during the first 3 days after birth were significantly higher in extremely premature neonates than in moderately premature and term neonates. Although the extremely premature gut contains relatively few mature enterocytes, I-FABP can be released more easily than in the more mature intestine because enterocyte cell membranes might be more vulnerable to insults. In addition, Corvaglia et al. demonstrated that splanchnic oxygenation measured at first feed by near-infrared spectroscopy was significantly lower in infants who later developed feeding intolerance [24]. They speculated that lower splanchnic oxygenation was related to a specific impairment of mesenteric hemodynamics or reflected poor intestinal growth and perfusion during fetal life and led to gastrointestinal complications. Kokesova A et al. also demonstrated that urinary I-FABP in gastroschisis patients, which might be released by increased intra-abdominal pressure, oxidative stress, and an increase in apoptotic activity of enterocytes, can be a biomarker for intestinal mucosal disruption [25]. In this study, younger gestational age and lower cord pH were the most consistent risk factors for increased levels of early I-FABP/creatinine_u_ in the multiple linear regression analysis. Based on these results, we can postulate that a more premature gut and acute perinatal ischemia might present with increased I-FABP levels early in the postnatal period.

We also investigated the correlations between early I-FABP/creatinine_u_ levels and neonatal morbidities in the normal group. BPD and ROP, which required laser surgery, were significantly correlated with early I-FABP/creatinine_u_ levels, and their statistical significance remained even after adjusting for gestational age. Although we identified the correlation between early I-FABP/creatinine_u_ levels and ROP and BPD, there have been no previous studies on the role of I-FABP in ROP and BPD. Although we identified the correlation between early I-FABP/creatinine_u_ levels and ROP and BPD, we could not determine its clinical significance because of the lack of studies. Further studies with a large population are needed to determine the significance of I-FABP for predicting major neonatal morbidity and long-term outcomes in preterm infants.

Our results also demonstrated significantly higher I-FABP/creatinine_u_ levels at 48 h after birth in patients with MOP than in the normal group but similar levels at 12 and 24 h after birth. Although the pathophysiology of MOP is poorly understood, because the pathophysiology of feeding intolerance is similar, prenatal intestinal hypoperfusion can contribute to delayed passage of meconium and the development of MOP [26,27,28,29]. The fact that I-FABP/creatinine_u_ levels increased at 48 h after birth in the MOP group might indicate the progression of intestinal injury after birth, such as aggravation of bowel dilatation and the progression of ischemia after feeding. Because of the small number of NEC patients, there were no statistically significant differences in early I-FABP/creatinine_u_ levels between the NEC group and the normal group.

This study has several strengths. We conducted a prospective cohort study of preterm infants to consistently collect data and considered many factors in the study design and data analysis and interpretation. We used urine samples for analyses. I-FABP is readily excreted by the kidneys and can be measured in urine within hours of tissue damage [30]. Urine is easy to collect in a large quantity with noninvasive procedures instead of measuring serial serum I-FABP levels. Several previous studies have shown that the urinary concentrations of some biomarkers depend on gestational age in preterm infants with kidney maturity. This may be due to the inability of immature tubules to reabsorb these proteins in underdeveloped kidneys. Therefore, we used the urine creatinine concentration to control kidney function. The advantage of analyzing collected urine during 48 h after birth is that we can directly evaluate the potential risk of prenatal and acute perinatal events.

This study has several limitations because of the small sample size, and the possible risks of underestimating the urine parameters due to the method of urine sample collection without cooling or extracting the samples for a long time. Further large-scale studies are needed to confirm the results of this study.

## 5. Conclusions

In conclusion, early urinary I-FABP/creatinine_u_ levels shortly after birth were correlated with cord pH and inversely correlated with gestational age and birth weight at birth in preterm infants. More immature guts and acute perinatal ischemia independently contributed to increased I-FABP levels shortly after birth. Because of the small sample size, further large-scale studies are needed.

## Figures and Tables

**Figure 1 children-08-00842-f001:**
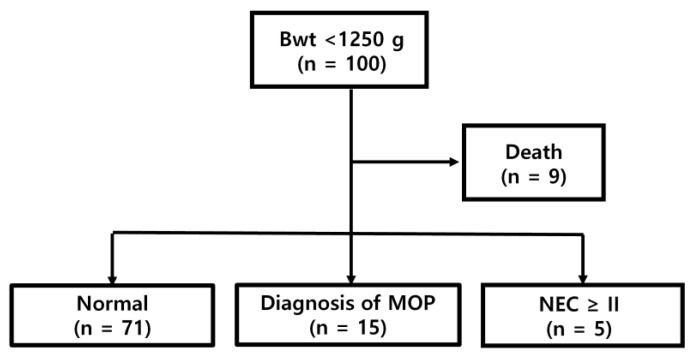
Flowchart of patients. Bwt, birth weight; MOP, meconium obstruction of prematurity; NEC, necrotizing enterocolitis.

**Figure 2 children-08-00842-f002:**
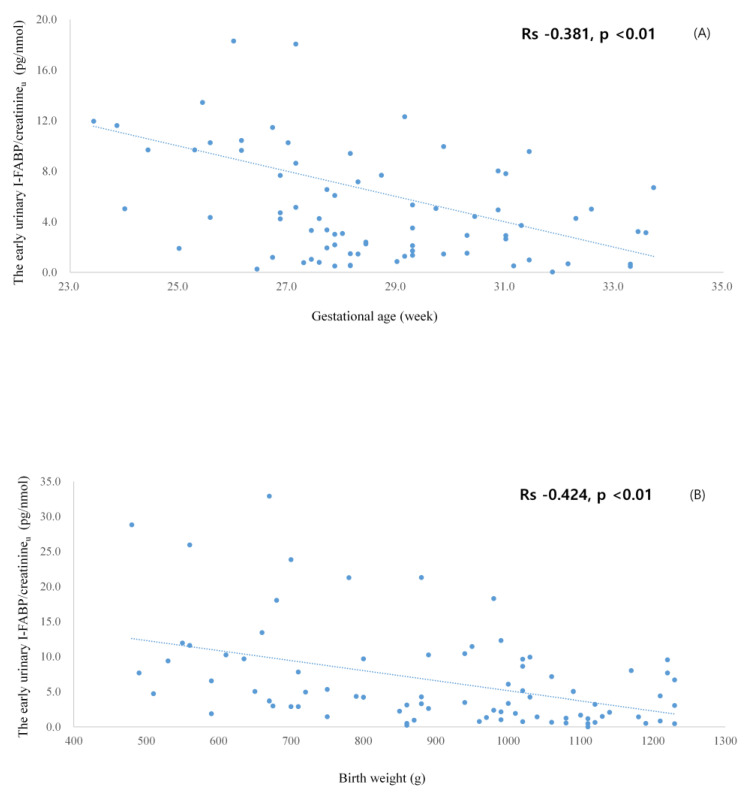
The correlations between early urinary I-FABP/Cr levels and gestational age (**A**) and birth weight (**B**). I-FABP, intestinal fatty acid binding protein; Cr, creatinine.

**Figure 3 children-08-00842-f003:**
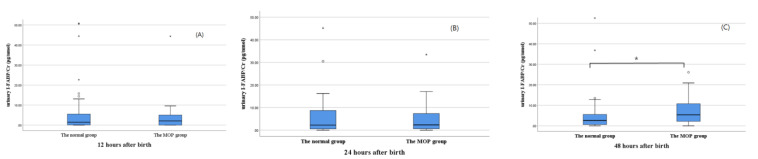
The comparison of urinary I-FABP/Cr levels between the control group and the MOP group at 12 h (**A**), 24 h (**B**), and 48 h (**C**) after birth. (* *p* = 0.025). I-FABP, intestinal fatty acid binding protein; Cr, creatinine; MOP, meconium of prematurity.

**Table 1 children-08-00842-t001:** Demographics and clinical characteristics of the study population.

	N = 100
Male/female	45/55
Gestational age (week)	28.1 ± 2.7 (23.4, 33.7)
Birth weight (g)	874.6 ± 221.1 (480, 1230)
Apgar score at 1 min	4.0 ± 2.1
Apgar score at 5 min	6.2 ± 1.9
Small for gestational age (<10 percentile)	31 (31)
Histologic chorioamnionitis	41 (41)
Oligohydramnios	25 (25)
PROM	28 (28)
Antenatal steroid	70 (70)
PIH	20 (20)
Gestational diabetes mellitus	4 (4)
Treated patent ductus arteriosus	49 (49)
Intraventricular hemorrhage (≥II)	17 (17)
Periventricular leukomalacia	7 (7)
Culture proven sepsis	20 (20)
Moderate or severe BPD	40 (40)
Necrotizing enterocolitis	5 (5.5)
MOP	15 (16.5)
TPN duration	19.1 ± 15.2
Admission duration (day)	81.9 ± 37.8
Mortality	9 (9)
I-FABP/Cr at 12 h after birth (pg/nmol)	5.25 ± 10.18
I-FABP/Cr at 24 h after birth (pg/nmol)	5.34 ± 7.53
I-FABP/Cr at 48 h after birth (pg/nmol)	5.20 ± 7.74

PROM, premature rupture of membrane; PIH, pregnancy-induced hypertension; BPD, bronchopulmonary dysplasia; MOP, meconium obstruction of prematurity; TPN, total parenteral nutrition; I-FABP, intestinal fatty acid binding protein; Cr, creatinine. Values are presented as the means ± SDs (minimum, maximum), or numbers (%).

**Table 2 children-08-00842-t002:** The correlations between early urinary I-FABP/urinary Cr levels and perinatal factors.

Associated Factors ^†^	rs	*p*-Value
1 min Apgar score	0.011	0.932
5 min Apgar score	−0.043	0.706
Cord pH	−0.436	<0.001
Base excess	−0.258	0.021

^†^ Analyzed by partial correlation analysis; all factors were adjusted by gestational age.

**Table 3 children-08-00842-t003:** The associations between early urinary I-FABP/urinary Cr levels and prenatal and postnatal factors.

Factors ^†^	*p*-Value
Pregnancy-induced hypertension	0.245
Histologic chorioamnionitis	0.892
Oligohydramnios	0.003 *
Premature rupture of membrane	0.170
Small for gestational age (<10%)	0.922
Intraventricular hemorrhage Grade ≥2	0.319
Periventricular leukomalacia	0.671
Culture proven sepsis	0.841
Bronchopulmonary dysplasia	0.023 *
Retinopathy of prematurity	0.025 *

* *p* < 0.05, ^†^ Analyzed by the Mann–Whitney U test. All factors were adjusted by gestational age.

**Table 4 children-08-00842-t004:** Multiple linear regression analyses on early urinary I-FABP/urinary Cr levels and perinatal factors.

Factors	B (95% CI)	*p*-Value
Gestational age	−0.131 (−0.214, −0.049)	0.002
Birth weight	−0.014 (−0.020, 0.007)	<0.001
Cord pH	−24.057 (−38.016, −10.099)	0.001
Base excess	−0.235 (−0.539, −0.069)	0.127

B, unstandardized coefficient; CI, confidence interval.

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
