# Peer review of "The Physiologic Significance of Early Urinary Intestinal Fatty Acid Binding Protein Levels in Preterm Infants: A Prospective Cohort Study"

_children, 2021, doi:10.3390/children8100842_

Round 1

Reviewer 1 Report

Congratulations to authors on the completion of this study. The study highlights some important findings and highlight the association of prematurity and I-FABP/creatinine levels.

  • The exclusion criteria in line 56 is not clear. "Severe congenital anomalies?". Were cyanotic congenital heart disease patients excluded? What about respiratory status of patients, intubated vs CPAP and their correlation with the I-FABP levels?
  • The inference in the discussion (and also in conclusion) in line 236 "According to our results, early urinary I-FABP levels might be an excellent biomarker for indicating intestinal ischemia, but the way fetal inflammatory status affects these levels 237 cannot be deduced from our study" can not be derived from the results of the current study. The results show that I-FABP levels are associated with lower GA and BW and pH but there is no direct assessemnt of "ischemia". Furthermore, there was no correlation with NEC with I-FABP levels so I do not think we can conclude I-FABP levels are associated with ischemia. 
  • Does age affect I-FABP levels? perhaps a better way to assess association of levels with NEC would have been to check levels in known NEC patients with normal controls of same GA and chronologic age to compare if there was a signficant difference. 

Author Response

  1. The exclusion criteria in line 56 is not clear. "Severe congenital anomalies?". Were cyanotic congenital heart disease patients excluded? What about respiratory status of patients, intubated vs CPAP and their correlation with the I-FABP levels?

In this study, we excluded major congenital anomalies;

  • congenital heart diseases which are responsible for mesenteric hypoxemia including cyanotic heart disease and ductus dependent congenital heart disease.
  • abdominal congenital anomalies, such as esophageal atresia, intestinal obstruction imperforated anus, gastroschisis/omphalocele,
  • other major anomalies including renal agenesis and neural tube defects
  • chromosomal anomalies.

We added more details to the exclusion criteria.

“. Patients with major congenital anomalies such as cyanotic heart diseases, ductus dependent heart disease, abdominal congenital anomalies, neural tube defect, and chromosomal anomalies, were excluded.”

In the present study, we collected initial urine samples at 12 ± 6, 24 ± 6, and 48 ± 6 hours after birth and we cannot separate urine samples into day 1 (within 24 hours) and day 2 (within 48 hours), and respiratory status of each patient changed during the first 48 hours.

The correlations between the respiratory status and the urinary I-FABP/urinary Cr levels at 24 and 48 hours after birth are shown in the following table.

Day 1 respiratory status

Non-invasive respiratory support (n=30)

Invasive respiratory support (n=61)

p-value

Urinary I-FABP/urine Cr at 12hours after birth (pg/nmol)

5.87 ± 10.06

4.94 ± 10.31

0.684

Urinary I-FABP/urine Cr at 24hours after birth (pg/nmol)

5.44 ± 6.62

5.29 ± 7.99

0.927

Day 2 respiratory status

Non-invasive respiratory support (n=38)

Invasive respiratory support (n=53)

p-value

Urinary I-FABP/urine Cr at 24hours after birth (pg/nmol)

5.99 ± 8.99

4.87 ± 6.33

0.509

Urinary I-FABP/urine Cr at 48hours after birth (pg/nmol)

5.33 ± 9.39

5.11 ± 6.38

0.902

Regarding respiratory status, the mean urinary I-FABP/urine Cr levels at 12 hours, 24 hours, and 48 hours after birth were not significantly different between the invasive respiratory support status and the noninvasive respiratory support status.

  1. The inference in the discussion (and also in conclusion) in line 236 "According to our results, early urinary I-FABP levels might be an excellent biomarker for indicating intestinal ischemia, but the way fetal inflammatory status affects these levels 237 cannot be deduced from our study" can not be derived from the results of the current study. The results show that I-FABP levels are associated with lower GA and BW and pH but there is no direct assessemnt of "ischemia". Furthermore, there was no correlation with NEC with I-FABP levels so I do not think we can conclude I-FABP levels are associated with ischemia. 

As you mentioned, we could not investigate the direct correlation with perinatal ischemia in this study. The present study demonstrated that early urinary I-FABP levels are relatively associated with lower umbilical pH, which might reflect perinatal distress. We agree to modify the sentence as you mentioned. We modified the sentence that ”an candidate biomarker for indicating intestinal ischemia”.

  1. Does age affect I-FABP levels? perhaps a better way to assess association of levels with NEC would have been to check levels in known NEC patients with normal controls of same GA and chronologic age to compare if there was a signficant difference. 

In this prospective cohort study, our primary aim was to determine the clinical meanings of early urinary I-FABP levels. We only compared the early I-FABP levels during 48 hours after birth between the normal group and the NEC group. As you mentioned, postnatal age and postmenstrual age should be considered and be adjusted to assess the exact associations of urinary I-FABP levels with NEC. Unfortunately, we did not collect the serial urine samples in the normal group. Therefore, we could not evaluate the correlations of urinary I-FABP levels with chronologic age, and cannot make comparisons on the urinary I-FABP levels between two groups at the exact time point which was at the first presentation of clinical symptom.

Further study which has a suitable design for the NEC is necessary.

Reviewer 2 Report

A good and well developed study which it is also well reported in the manuscript. I would suggest making clear the two main weaknesses of the study: The small sample size of the study (as indicated by the authors at the end of the discussion section, but should be stated in the abstract and conclusions) and the possible risks of underestimation of the urine parameters due to the method of collection without cooling the sample or extracting them for a long time.

Author Response

According to your recommendation, we added the weaknesses of this study in the abstract and the conclusion section as “because of small sample size, further large-scale studies are needed”. We also added more detailed description as “This study has several limitations because of the small sample size, and the possible risks of underestimating the urine parameters due to the method of urine sample collection without cooling or extracting them for a long time”.

Reviewer 3 Report

The paper is well written and presents some interesting findings.

Author Response

none